rsos.royalsocietypublishing.org

# Crystal growth of $La_{2/3-x}Li_{3x}TiO_3$ by the TSFZ method

Yuki Maruyama, Shiho Minamimure, Chinatsu Kobayashi, Masanori Nagao, Satoshi Watauchi and Isao Tanaka

Center for Crystal Science and Technology, University of Yamanashi, 7-32 Miyamae, Kofu, Yamanashi 400-8511, Japan

 IT, 0000-0002-2736-7107

environmental engineering/crystal engineering/ materials science

crystal growth, travelling solvent floating zone method, anisotropic ionic conductivity, Li ion battery

**Author for correspondence:**
Isao Tanaka
e-mail: itanaka@yamanashi.ac.jp

Double-perovskite-type $La_{2/3-x}Li_{3x}TiO_3$ (LLT) crystals were grown by the travelling solvent floating zone (TSFZ) method. When the floating zone (FZ) crystal growth method was applied, the $La_2Ti_2O_7$ phase was deposited as an inclusion in the initial growth region. Using the TSFZ crystal growth method, however, inclusion-free LLT crystals were obtained for a 10 mol% $La_2Ti_2O_7$-poor composition solvent relative to the stoichiometric LLT crystals. The molten zone was initially unstable as a result of habit plane formation during the crystal growth; however, the molten zone was stably maintained for a long period of time by decreasing the feed rate compared with the growth rate. Hence, LLT crystals of approximately 5 mm$\varphi$ and 37 mm in length were obtained. The anisotropic ionic conductivity of the crystals annealed in air was $\sigma[110]/\sigma[001] \approx 3$, with $\sigma[110] = 1.64 \times 10^{-3}\,S\,cm^{-1}$ and $\sigma[001] = 5.26 \times 10^{-4}\,S\,cm^{-1}$. LLT single crystals are candidates for high-performance solid-state electrolytes in all-solid-state Li ion batteries.

This article has been edited by the Royal Society of Chemistry, including the commissioning, peer review process and editorial aspects up to the point of acceptance.

## 1. Introduction

Conventional Li ion batteries incorporating liquid electrolytes are widely used in electric and portable devices; however, these batteries are associated with various potential problems such as electrolyte leakage and fire. By contrast, all-solid-state Li ion batteries with a solid electrolyte are highly safe, being non-flammable and having zero leakage. Moreover, all-solid-state Li ion batteries offer significant advantages over conventional batteries, such as thermal stability and large potential windows allowing the use of high-voltage cathode materials and/or metallic Li anodes [1–3]. Thus, all-solid-state Li ion batteries are promising as next-generation storage batteries.

Inorganic solid electrolytes comprise sulfide and oxide solid electrolytes. Sulfide solid electrolytes are reported to possess high ionic conductivities ($\sigma$) of $10^{-2}\,S\,cm^{-1}$ comparable to those

rsos.royalsocietypublishing.org R. Soc. open sci. 5: 181145

of liquid electrolytes [4]. However, sulfide solid electrolytes are reactive to moisture in the atmosphere. Oxide solid electrolytes such as perovskite structures, e.g. $La_{2/3-x}Li_{3x}TiO_3$ (LLT) [5–8]; Na super ionic conductor (NASICON) structures, e.g. $Li_{1+x}Al_xTi_{2-x}(PO_4)_3$ [9,10]; and garnet electrolytes, e.g. $Li_7La_3Zr_2O_{12}$ [11,12], have been extensively studied because of their high chemical stability and high $\sigma$. It has been also reported that the $\sigma$ of perovskite-type $ABO_3$ is improved by the substitution of other elements on the A- or B-sites, such as $Ln_{1/2}Li_{1/2}TiO_3$ (Ln = La, Pr, Nd, Sm) [13], $R_{1/4}Li_{1/4}TaO_3$ (R = La, Nd, Sm, Y) [14,15] and $LiSr_{1.65}M_{1.3}M'_{1.7}O_9$ (M = Ti Zr, M' = Nb, Ta) [16–18]. Among them, LLT is a promising electrolyte because of its high $\sigma$ [5].

It is known that the LLT crystal structure depends on the Li concentration [7,19]. LLT ($x = 0.167$) is a cubic perovskite structure (space group: $Pm3m$), whereas LLT ($0.03 \leqq x \leqq 0.167$) is a tetragonal double-perovskite structure (space group: $P4/mmm$). Double-perovskite-type LLT is composed of alternately stacked La-rich and -poor layers along the $c$-axis [20]. The $\sigma$ found in LLT ($x = 0.117$), i.e. approximately $10^{-3}$ S cm$^{-1}$ at room temperature, is the highest yet reported for Li concentrations [6,7]. Thus, double-perovskite-type LLT has been identified as having the most promising structure for use in an Li ion conductor.

To date, a sintered body of the ionic conductor has been used for oxide solid electrolytes. However, the $\sigma$ of a solid electrolyte obtained using powder decreases with the presence of grain boundary resistance [21,22]. By contrast, a solid electrolyte based on a single crystal has no grain boundary and high $\sigma$ is expected. To advance this field, LLT single crystals are required for experiment to clarify their basic physical properties (e.g. their anisotropic ionic conductivity) and to apply a substrate for application in high-performance all-solid-state batteries. Inaguma *et al.* reported the growth of LLT single crystals with dimensions of $3 \times 2 \times 1$ mm$^3$ using the floating zone (FZ) method, and discussed their $\sigma$ values [23]. Moreover, Varez *et al.* have reported the growth of LLT fibre single crystals using the laser FZ method [24]. However, large single crystals have not yet been obtained. The end $La_2Ti_3O_9$, i.e. the end member of $La_{2/3-x}Li_{3x}TiO_3$, melts incongruently to $La_2Ti_2O_7$ and a liquid at $1660°C$, according to the phase diagram of the $La_2O_3$-$TiO_2$ system [25]. It is likely that LLT is also an incongruent melting compound. The travelling solvent floating zone (TSFZ) method, which involves FZ growth using a solvent, is a powerful crystal growth technique for an incongruent-melting compound and solid solutions such as $Y_3Fe_5O_{12}$ and $La_{2-x}Sr_xCuO_4$ superconductors [26–28].

In the present study, the incongruent-melting behaviour of LLT is confirmed using the FZ method, and larger LLT single crystals are grown by the TSFZ method. The solvent composition for TSFZ growth is determined by clarifying the inclusion phases in crystals grown by the FZ method. The TSFZ growth conditions, such as the solvent composition and feed rate, are then optimized for large LLT single-crystal growth. Finally, the anisotropic ionic conductivity of the LLT grown crystals is clarified.

# 2. Experimental procedure

$La_2O_3$ (Rare Metallic Co., Ltd; 99.99%), $Li_2CO_3$ (Rare Metallic Co., Ltd; 99.99%) and $TiO_2$ (Rare Metallic Co., Ltd; 99.99%) were used as starting materials, being weighed at the $La_{0.55}Li_{0.35}TiO_3$ ($x = 0.117$) stoichiometric composition and mixed with ethanol. The mixed powder was calcined at $800°C$ for 1 h in air. Then, the powder was ground, before being calcined at $1100°C$ for 12 h in air. The calcined powder was formed into a cylindrical shape by rubber pressing. The resultant rod was then sintered at $1300°C$ for 12 h in air and used as a feed rod for crystal growth. Solvents with $TiO_2$-excess or $La_2Ti_2O_7$-poor compositions relative to the LLT stoichiometric composition were prepared using the same procedure as feed rod preparation.

For crystal growth, an infrared convergent heating image furnace (Crystal Systems Corporation, FZ-T-4000-H) equipped with four 300 W halogen lamps was used. The crystal growth was conducted under Ar gas flow conditions. The feed and growth rates were 3–5 and 5 mm h$^{-1}$, respectively. The rotation rates of the feed rod and the grown crystal were 8 and 40 r.p.m., respectively. The grown crystals were annealed at $1000°C$ for 20–30 h under the oxygen gas flow.

The precipitated phases in the grown crystals were observed using an electron probe micro-analyser (EPMA, JEOL JXA-8200). The crystals were cut along the growth direction and the cut surfaces were mirror-polished. Elemental mapping images of La and Ti in the mirror-polished samples were measured using the EPMA. Also, the concentrations of La and Ti in the grown crystals were determined by a quantitative analysis using a $La_2Ti_2O_7$ single crystal as a standard sample, and Li concentration $3x$ in the composition $La_{2/3-x}Li_{3x}TiO_3$ was estimated using the atomic ratio La/Ti. The crystallographic axis was identified using the back-reflection Laue X-ray diffraction method and X-ray

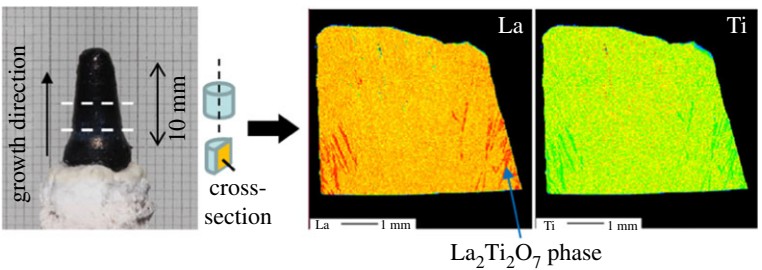

rsos.royalsocietypublishing.org    R. Soc. open sci. **5**: 181445

**Figure 1.** Photograph of crystal grown by the FZ method and mapping images of La and Ti within crystals.

diffraction (XRD, Bruker Discover D8 with a 2D detector). The Li ion conductivity of the samples was measured by an AC complex impedance method. Au-Pd thin film was sputtered on both sides of the sample surface to maintain the electrode ohmic contact. The complex impedance was measured using an inductance-capacitance-resistance (LCR) meter (Iwatsu Electric Co., Ltd; PSM 1700) in the frequency range of 1 Hz to 10 MHz at room temperature.

# 3. Results and discussion

## 3.1. Crystal growth by the floating zone method

Figure 1 presents a photograph of a crystal grown by the FZ method, along with mapping images of the cross-section of the grown crystal. Crystals of approximately 4 mm$\varphi$ and 11 mm in length were obtained. The crystals were black because of oxygen deficiency due to the growth in Ar gas. During the application of the FZ crystal growth method, it was difficult to maintain a stable molten zone because of inhomogeneous melting of the feed rod. The EPMA analysis indicated that the $La_2Ti_2O_7$ phase was deposited as an inclusion in the initial growth region. This result indicates that LLT is an incongruently melting compound, because the $La_2Ti_2O_7$ phase was deposited as a primary phase from the LLT melt. By contrast, the TSFZ method was expected to be effective for LLT crystal growth as it can suppress deposition of $La_2Ti_2O_7$ inclusions and decrease the growth temperature to reduce Li evaporation from the molten zone.

## 3.2. Solvent composition optimization for travelling solvent floating zone growth

During crystal growth by the FZ method, the LLT was found to melt incongruently to the $La_2Ti_2O_7$ phase and a liquid. In the next stage of this study, we attempted growth of LLT crystals by the TSFZ method, using solvents with $TiO_2$-excess or $La_2Ti_2O_7$-poor composition relative to the LLT stoichiometric composition. From the phase diagram of the $TiO_2-La_2O_3$ system [25], the composition of the liquid coexisting with the $La_2Ti_3O_9$ phase could be $TiO_2$-excess compared to $La_2Ti_3O_9$, as the $La_2Ti_2O_7$ composition is $TiO_2$-poor compared to $La_2Ti_3O_9$, the end member of LLT ($x = 0.00$). First, TSFZ growth was performed using a solvent with 3–6 mol% $TiO_2$-excess composition relative to the $La_2Ti_3O_9$ composition. The $La_2Ti_2O_7$ phase was detected in the grown crystals when the $TiO_2$-excess composition solvent was used, and deposition of the $La_2Ti_2O_7$ phase was not suppressed. Next, 5 or 10 mol% $La_2Ti_2O_7$-poor composition solvents relative to the $La_{2/3-x}Li_{3x}TiO_3$ ($x = 0.117$) composition were used. For those solvents, figure 2 shows photographs of the as-grown crystals and concentration mapping images of La and Ti in the cross-section parallel to the growth direction. Compared with the FZ growth case, the crystal growth stabilized when the solvent was used. Hence, LLT crystals of approximately 5 mm$\varphi$ and 20 mm in length were obtained. The concentration mapping image of the crystal obtained using the 5 mol% $La_2Ti_2O_7$-poor solvent shown in figure 2a indicates that the $La_2Ti_2O_7$ phase was deposited as an inclusion in the initial growth region. By contrast, the $La_2Ti_2O_7$ phase was not detected in the crystal grown using the 10 mol% $La_2Ti_2O_7$-poor solvent, as shown in figure 2b. Therefore, inclusion-free LLT crystals were grown using the solvent with 10 mol% $La_2Ti_2O_7$-poor composition relative to the stoichiometric LLT ($x = 0.117$).

## 3.3. Molten zone stabilization through feeding and growth rate control

Once the habit planes were formed, the melt in the molten zone tended to spill outward, because of the fluctuation of the mass balance between the feed and crystallization volumes. To maintain the molten zone stability for a long period of time, the balance between the feed and growth rates was investigated.

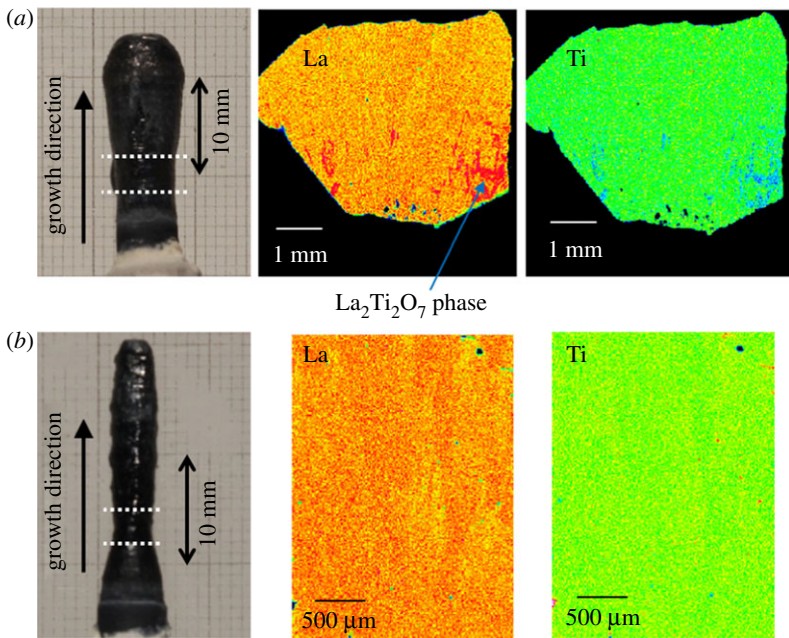

**Figure 2.** Photographs and La and Ti concentration mapping images for crystals grown by the TSFZ method with (*a*) 5 and (*b*) 10 mol% $La_2Ti_2O_7$-poor solvent.

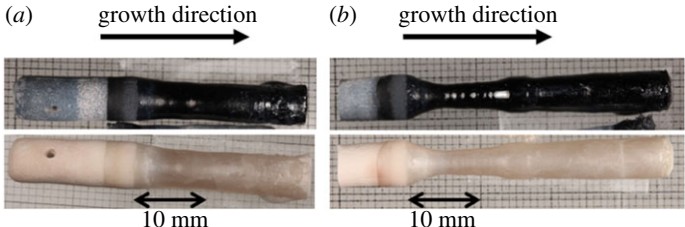

**Figure 3.** Grown (upper) and annealed (lower) crystals obtained at different feed rates: (*a*) 5 and (*b*) 3 mm h$^{-1}$.

To control the mass balance between feeding and crystallization, the feed rate was changed with respect to the 5 mm h$^{-1}$ growth rate. When the feed rate was identical to the 5 mm h$^{-1}$ growth rate, the molten zone volume became large, and the melt near the interface between the feed and crystal formed a bulge. It seems that a feed oversupply occurred, which limited the LLT crystal growth to 20 mm in length because of the melt drop. By contrast, a suitably large molten zone was stably maintained for a long period of time by decreasing the feed rate to 3 mm h$^{-1}$. Figure 3 shows the grown crystals and the crystals annealed in oxygen. LLT crystals of approximately 5 mm$\varphi$ and 37 mm in length were obtained by controlling the feed rate relative to the growth rate. Again, note that black crystals were obtained because of oxygen vacancy in the crystals, as the crystals were grown in an Ar gas flow. However, when the grown crystals were annealed in an oxygen gas flow, they became colourless. The chemical composition of the grown crystals was determined to be $La_{0.61\pm0.02}Li_{0.17\pm0.006}TiO_3$ ($x = 0.057 \pm 0.002$) by quantitative analysis using EPMA. The Li content in the grown crystals was lower than that in the feed. The low Li content is due to the evaporation of Li from the molten zone during the crystal growth and a lower distribution coefficient of Li into LLT. Figure 4 shows the XRD patterns of the LLT feed rod sintered at 1300°C for 12 h in air and the grown crystals. The XRD pattern of the grown crystals is attributed to a tetragonal double-perovskite-type LLT. Thus, we successfully obtained double-perovskite-type LLT crystals by the TSFZ method. The diffraction peaks of the grown crystals shifted to lower angle than those of the feed rod. It had been reported that the LLT lattice expands with the extra Li evaporation [21]. The peak shift in the grown crystals agrees with the results of the quantitative analysis by EPMA. The diffraction intensity due to the superstructure in the grown crystals was high compared with that of the sintered LLT crystals. This indicates that the grown crystals have a high crystallinity. Cracks due to annealing were observed in those crystals. In future work, crack-free LLT crystal growth is expected through optimization of the growth conditions, such as the Li concentration and growth rate.

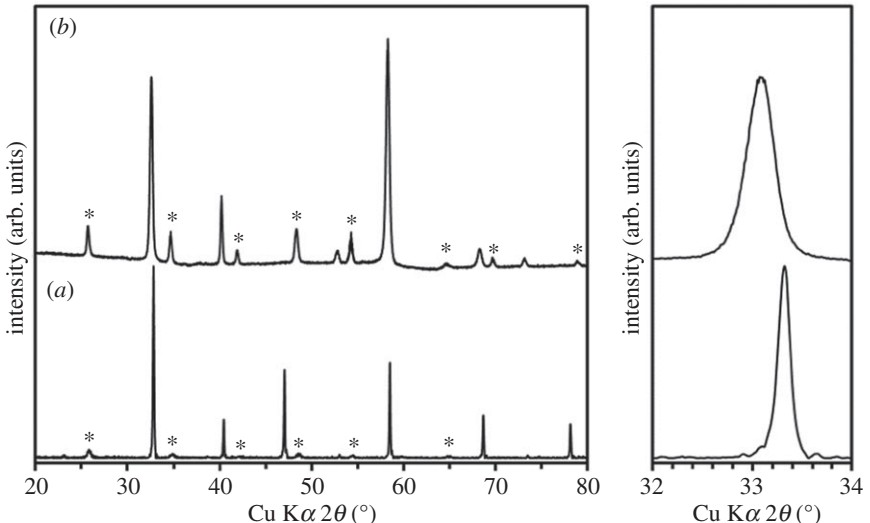

**Figure 4.** XRD patterns of (*a*) the feed rod sintered at 1300°C for 12 h in air and (*b*) the grown crystals. The asterisks * in the XRD patterns indicate the diffraction peaks corresponding to a superstructure in the double-perovskite structure of LLT.

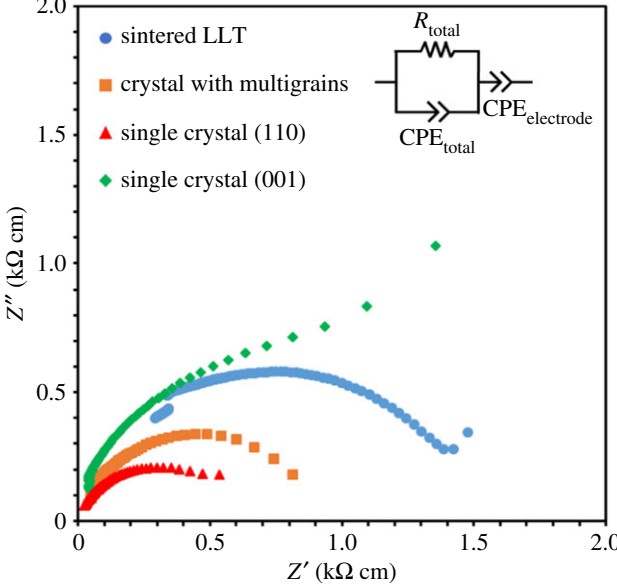

**Figure 5.** Impedance plots of LLT samples at room temperature. The crystals were annealed in oxygen.

The crystallographic axes were identified by the back-reflection Laue X-ray diffraction method and X-ray diffraction analysis was performed with a 2D detector. The growth direction and habit planes were [001] and (110), respectively.

## 3.4. Ionic conductivity of LLT crystals

Figure 5 shows the complex AC impedance plots for [110] and [001] in the LLT single crystals, sintered LLT and an LLT crystal with multigrains at room temperature, where the crystal samples annealed in oxygen were used for the measurement. A semicircular trend is observed at a higher frequency range, whereas a straight line is apparent at a lower frequency range. It is difficult to separate the bulk resistance and grain boundary resistance because only one semicircle is observed. In the impedance plots, the semicircle represents the total resistance of the solid electrolyte. The straight line is attributable to the electrode polarization in the blocking electrodes. The total resistance of the sample was estimated from the crossing point of the extrapolated semicircle with the real axis in the lower frequency range.

**Table 1.** Ionic conductivities of LLT samples at room temperature.

| sample | ionic conductivity $\sigma$ (S cm$^{-1}$) |
| --- | --- |
| single crystal [110] | $1.64 \times 10^{-3}$ |
| single crystal [001] | $5.26 \times 10^{-4}$ |
| crystal with multigrains | $1.18 \times 10^{-3}$ |
| sintered LLT | $6.90 \times 10^{-4}$ |

The $\sigma$ values of the samples are listed in table 1. The ionic conductivities along [110] and [001] in the LLT single crystals were $\sigma[110] = 1.64 \times 10^{-3}$ S cm$^{-1}$ and $\sigma[001] = 5.26 \times 10^{-4}$ S cm$^{-1}$, respectively. The anisotropic ionic conductivity of the grown crystal was $\sigma[110]/\sigma[001] \approx 3$. It has been reported that double-perovskite-type LLT is composed of alternately stacked La-rich and -poor layers [20]. A large amount of Li$^{+}$ occupies the La$^{3+}$ sites in the La-poor layer [20], and the Li$^{+}$ ions can move quickly within the La-poor layer because of the high Li$^{+}$ concentration. However, the Li$^{+}$ migration tends to be blocked by La$^{3+}$ cations in the La-rich layer [20,29], suggesting that Li$^{+}$ ions can more easily migrate along [110] than [001]. The $\sigma$ in the sintered LLT was $6.90 \times 10^{-4}$ S cm$^{-1}$, slightly higher than that along [001] in the LLT single crystals. This finding suggests that the mobility of Li$^{+}$ along [001] in LLT is a determining factor of the $\sigma$ in LLT. The $\sigma$ in the LLT crystal with multigrains was $1.18 \times 10^{-3}$ S cm$^{-1}$, twice that of the sintered LLT. Thus, a high $\sigma$ of $10^{-3}$ S cm$^{-1}$ was obtained in the crystals grown by the TSFZ method. We are planning to investigate the relationship between the Li content of the grown crystals and their $\sigma$ values in future work, by considering the Li concentration within the LLT crystals.

# 4. Conclusion

In this study, LLT was confirmed to be an incongruent-melting compound and larger LLT single crystals were grown by the TSFZ method. The incongruent melting of LLT was confirmed because the La$_2$Ti$_2$O$_7$ phase was deposited as a primary phase in the initial growth region during FZ growth of LLT crystals. Although LLT crystals of approximately 4 mm$\varphi$ and 11 mm in length were obtained through the FZ method, it was difficult to maintain the stable molten zone for a long period of time because of the inhomogeneous dissolution of the feed rod. For TSFZ growth of LLT, inclusion-free LLT single crystals were obtained when a solvent of 10 mol% La$_2$Ti$_2$O$_7$-poor composition relative to the stoichiometric LLT was used. By contrast, a solvent with 5 mol% La$_2$Ti$_2$O$_7$-poor composition relative to the stoichiometric LLT caused precipitation of the La$_2$Ti$_2$O$_7$ phase as an inclusion in the grown crystals. Moreover, the molten zone was stably maintained for long period of time by decreasing the feed rate from 5 to 3 mm h$^{-1}$. Hence, LLT crystals of approximately 5 mm$\varphi$ and 37 mm in length were obtained by the TSFZ method. Anisotropic ionic conductivity of $\sigma[110]/\sigma[001] \approx 3$ at room temperature was observed in the LLT single crystals, with $\sigma[110] = 1.64 \times 10^{-3}$ S cm$^{-1}$ and $\sigma[001] = 5.26 \times 10^{-4}$ S cm$^{-1}$. LLT single crystals orientated parallel to the $c$-axis are expected to be candidates for high-performance solid-state electrolytes in all-solid-state Li ion batteries.

Ethics. We have already received ethical approval from APRIN e-learning program in Japan.
Data accessibility. We have already included all data within the manuscript.
Authors' contributions. S.M., C.K., M.N. and S.W. carried out crystal growth of LLT and characterization of the grown crystals. Y.M. carried out measurement of ionic conductivity, participated in data analysis and drafted the manuscript. I.T. conceived of the study, designed the study, coordinated the study and helped draft the manuscript. All authors gave final approval for publication.
Competing interests. We have no competing interests.
Funding. The source of our funding is as follows. Research budget supported by University of Yamanashi in Japan, grant no. 11101004210500000, and JSPS KAKENHI in Japan, grant no. JP16K05930.

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
