## [Reviewer comments · Royal Society Open Science]

Review History

RSOS-181445.R0 (Original submission)

Review form: Reviewer 1 (Surjeet Singh)

Is the manuscript scientifically sound in its present form?

Yes

Are the interpretations and conclusions justified by the results?

Yes

Is the language acceptable?

Yes

Is it clear how to access all supporting data?

Yes

Do you have any ethical concerns with this paper?

No

Have you any concerns about statistical analyses in this paper?

No

Recommendation?

Accept with minor revision (please list in comments)

Comments to the Author(s)

The paper describe the growth of (La, Li)TiO₃ using the TSFZ method. LLT is a potential battery material, and therefore I consider the present work timely and worth publishing. However, there are few important points that authors should consider including in the manuscript to make it more credible and useful:

1. Powder x-ray diffraction pattern of the successfully grown crystals (in Fig. 4) should be included to confirm their crystal structure (this is important since the structure is governed by the final Li concentration, and Li being volatile the final composition may differ from the starting composition).
2. EDX or EPMA compositions (average value and standard deviation) of the successfully grown crystals should be included.
3. It would be nice if AC conductivity of oxygen annealed crystals is also included.
4. Habit plane should be defined. It is not clear from the description given in paragraph 1 of section 3.3 as to how the habit plane formation hinders the growth. However, the paragraph 2 is clear where feeding speed is discussed. I will appreciate if paragraph 1 is also made more useful.
5. Figure 3 is not useful as it doesn't reveal much. You can however choose a picture showing the habit planes. If such a picture is not available you can remove Fig. 3, as you will anyway have one extra figure showing the powder XRD of the crystals.
6. No mention of Li evaporation during the growth? In my view, this is an important parameter for reproducibility of these experiments. I suggest that this point be mentioned and discussed.

Review form: Reviewer 2

Is the manuscript scientifically sound in its present form?

No

Are the interpretations and conclusions justified by the results?

Yes

Is the language acceptable?

Yes

Is it clear how to access all supporting data?

Not Applicable

Do you have any ethical concerns with this paper?

No

Have you any concerns about statistical analyses in this paper?

No

Recommendation?

Major revision is needed (please make suggestions in comments)

Comments to the Author(s)

Dear authors,

This paper describes crystal growth of LLT by TSFZ method. They paid a lot of attention to decomposition of the compound, incongruent melting, to achieve stable growth conditions. As a result, they succeeded to obtain really bulk single crystals. I think this is a very good work in terms of crystal growth.

To make this manuscript more interesting one for the readers, I encourage authors to add some results and descriptions.

Authors mentioned, 'decrease the growth temperature to prevent Li evaporation from the molten zone.' However, they did not disclose the effect of Li loss. It can be possible to estimate Li loss during growth by measuring total weight of feed rod and obtained crystals. I know not only loss of lithium but also oxygen deficiency is another cause of weight loss. However, rough estimation of Li loss may be possible.

I suppose Li-loss is also an important fact relating to the authors claim. In order to conclude incongruent melting, it is necessary to show that change of composition in molten zone was not due to loss of lithium but due to incongruent melting. To mention about phase diagram, chemical composition of the system must be kept constant during the process. Loss of lithium is a potential cause for misleading of phase relationships. I do not know if the authors have results to mention about Li loss during growth. However, I would like to ask the author to give quantitative or qualitative estimation of Li-loss during growth process.

The conductivity is affected by the number of carrier and mobility. It seems that anisotropy in conduction was more significant than the value reported in a literature. I suppose this is an evidence for high quality of the crystal grown by the authors. However, it is not possible to discuss conductivity in detail only from the results measured at room temperature. To give useful information to readers, it is a good idea to show temperature dependence of conductivity. Evaluation of the activation energy of conduction is necessary for understanding of the conduction mechanism. As the quality of the crystal seems to be sufficiently high to evaluate reliable physical parameters, I encourage the authors to work on temperature dependence measurements.

Decision letter (RSOS-181445.R0)

03-Oct-2018

Dear Professor Tanaka:

Title: Crystal growth of $\text{La}_{2/3-x}\text{Li}_{3x}\text{TiO}_3$ by TSFZ method

Manuscript ID: RSOS-181445

The editor assigned to your manuscript has now received comments from reviewers. We would

like you to revise your paper in accordance with the referee and Subject Editor suggestions which can be found below (not including confidential reports to the Editor). Please note this decision does not guarantee eventual acceptance.

Please submit your revised paper before 26-Oct-2018. Please note that the revision deadline will expire at 00.00am on this date. If we do not hear from you within this time then it will be assumed that the paper has been withdrawn. In exceptional circumstances, extensions may be possible if agreed with the Editorial Office in advance. We do not allow multiple rounds of revision so we urge you to make every effort to fully address all of the comments at this stage. If deemed necessary by the Editors, your manuscript will be sent back to one or more of the original reviewers for assessment. If the original reviewers are not available we may invite new reviewers.

Please also include the following statements alongside the other end statements. As we cannot publish your manuscript without these end statements included, if you feel that a given heading is not relevant to your paper, please nevertheless include the heading and explicitly state that it is not relevant to your work.

- Ethics statement

Please clarify whether you received ethical approval from a local ethics committee to carry out your study. If so please include details of this, including the name of the committee that gave consent in a Research Ethics section after your main text. Please also clarify whether you received informed consent for the participants to participate in the study and state this in your Research Ethics section.

OR

Please clarify whether you obtained the necessary licences and approvals from your institutional animal ethics committee before conducting your research. Please provide details of these licences and approvals in an Animal Ethics section after your main text.

OR

Please clarify whether you obtained the appropriate permissions and licences to conduct the fieldwork detailed in your study. Please provide details of these in your methods section.

- Data accessibility

It is a condition of publication that you make available the data and research materials supporting the results in the article. Datasets should be deposited in an appropriate publicly available repository and details of the associated accession number, link or DOI to the datasets must be included in the Data Accessibility section of the article (<http://royalsocietypublishing.org/instructions-authors#question17>). Reference(s) to datasets should also be included in the reference list of the article with DOIs (where available).

Please include a Data Availability section after your main text stating where supporting data are

available from, or where they will be made available should your article be accepted for publication.

If you wish to submit your supporting data or code to Dryad (<http://datadryad.org/>), or modify your current submission to dryad, please use the following link:
<http://datadryad.org/submit?journalID=RSOS&manu=RSOS-181445>

- **Competing interests**

Please include a Competing Interests section after your main text declaring any financial or non-financial competing interests. If you have no competing interests please state 'I/we have no competing interests.'

- **Authors' contributions**

Please include an Authors' Contributions section at the end of your main text detailing the contribution of each author. All authors should have read and approved the manuscript before submission and this should be stated in the Authors' Contributions section.

The list of Authors should meet all of the following criteria; 1) substantial contributions to conception and design, or acquisition of data, or analysis and interpretation of data; 2) drafting the article or revising it critically for important intellectual content; and 3) final approval of the version to be published.

- **Acknowledgements**

- **Funding statement**

Please include a funding section after your main text which lists the source of funding for each author.

Yours sincerely,
Dr Laura Smith, MRSC
Publishing Editor, Journals
Royal Society of Chemistry,
Thomas Graham House,
Science Park, Milton Road,
Cambridge, CB4 0WF, UK

Royal Society Open Science - Chemistry Editorial Office

RSC Associate Editor:
Comments to the Author:
(There are no comments.)

RSC Subject Editor:
Comments to the Author:
(There are no comments.)

Reviewers' Comments to Author:
Reviewer: 1

Comments to the Author(s)

The paper describe the growth of (La, Li)TiO₃ using the TSFZ method. LLT is a potential battery material, and therefore I consider the present work timely and worth publishing. However, there are few important points that authors should consider including in the manuscript to make it more credible and useful:

1. Powder x-ray diffraction pattern of the successfully grown crystals (in Fig. 4) should be included to confirm their crystal structure (this is important since the structure is governed by the final Li concentration, and Li being volatile the final composition may differ from the starting composition).
2. EDX or EPMA compositions (average value and standard deviation) of the successfully grown crystals should be included.
3. It would be nice if AC conductivity of oxygen annealed crystals is also included.
4. Habit plane should be defined. It is not clear from the description given in paragraph 1 of section 3.3 as to how the habit plane formation hinders the growth. However, the paragraph 2 is clear where feeding speed is discussed. I will appreciate if paragraph 1 is also made more useful.
5. Figure 3 is not useful as it doesn't reveal much. You can however choose a picture showing the habit planes. If such a picture is not available you can remove Fig. 3, as you will anyway have one extra figure showing the powder XRD of the crystals.
6. No mention of Li evaporation during the growth? In my view, this is an important parameter for reproducibility of these experiments. I suggest that this point be mentioned and discussed.

Reviewer: 2

Comments to the Author(s)
Dear authors,

This paper describes crystal growth of LLT by TSFZ method. They paid a lot of attention to decomposition of the compound, incongruent melting, to achieve stable growth conditions. As a result, they succeeded to obtain really bulk single crystals. I think this is a very good work in terms of crystal growth.

To make this manuscript more interesting one for the readers, I encourage authors to add some results and descriptions.

Authors mentioned, 'decrease the growth temperature to prevent Li evaporation from the molten zone.' However, they did not disclose the effect of Li loss. It can be possible to estimate Li loss during growth by measuring total weight of feed rod and obtained crystals. I know not only loss of lithium but also oxygen deficiency is another cause of weight loss. However, rough estimation of Li loss may be possible.

I suppose Li-loss is also an important fact relating to the authors claim. In order to conclude incongruent melting, it is necessary to show that change of composition in molten zone was not due to loss of lithium but due to incongruent melting. To mention about phase diagram, chemical composition of the system must be kept constant during the process. Loss of lithium is a potential cause for misleading of phase relationships. I do not know if the authors have results to mention about Li loss during growth. However, I would like to ask the author to give quantitative or qualitative estimation of Li-loss during growth process.

The conductivity is affected by the number of carrier and mobility. It seems that anisotropy in conduction was more significant than the value reported in a literature. I suppose this is an evidence for high quality of the crystal grown by the authors. However, it is not possible to discuss conductivity in detail only from the results measured at room temperature. To give useful information to readers, it is a good idea to show temperature dependence of conductivity. Evaluation of the activation energy of conduction is necessary for understanding of the conduction mechanism. As the quality of the crystal seems to be sufficiently high to evaluate reliable physical parameters, I encourage the authors to work on temperature dependence measurements.

Author's Response to Decision Letter for (RSOS-181445.R0)

See Appendix A.

Decision letter (RSOS-181445.R1)

06-Nov-2018

Dear Professor Tanaka:

Title: Crystal growth of $\text{La}_{2/3-x}\text{Li}_{3x}\text{TiO}_3$ by TSFZ method

Manuscript ID: RSOS-181445.R1

It is a pleasure to accept your manuscript in its current form for publication in Royal Society Open Science. The chemistry content of Royal Society Open Science is published in collaboration with the Royal Society of Chemistry.

RSC Associate Editor
Comments to the Author:
(There are no comments.)

Reviewer(s)' Comments to Author:

Appendix A

We thank referees for careful reading our manuscript and for giving useful comments.

We have revised the manuscript ID RSOS-181445 entitled " Crystal growth of $\text{La}_{2/3-x}\text{Li}_{3x}\text{TiO}_3$ by TSFZ method".

We are looking forward to publishing our manuscript in Royal Society Open Science.

Our responses to the referee's comments are as follows:

Reviewers' Comments to Author:

Reviewer: 1

Comments to the Author(s)

The paper describe the growth of (La, Li) TiO_3 using the TSFZ method. LLT is a potential battery material, and therefore I consider the present work timely and worth publishing. However, there are few important points that authors should consider including in the manuscript to make it more credible and useful:

1. Powder x-ray diffraction pattern of the successfully grown crystals (in Fig. 4) should be included to confirm their crystal structure (this is important since the structure is governed by the final Li concentration, and Li being volatile the final composition may differ from the starting composition).

→We thank the referee for fruitful suggestions. We included the XRD pattern of the grown crystal on Fig 4. We confirmed that the crystal structure of grown crystal is a tetragonal double-perovskite structure. We think that the final Li composition may differ from the starting composition. However, we obtained LLT crystal with a tetragonal double-perovskite structure.

We added following sentence in section 3.3:

Figure 4 shows the XRD patterns of the LLT feed rod sintered at 1300°C for 12 h in air and the grown crystals. The XRD pattern of the grown crystals is attributed to a tetragonal double-perovskite-type LLT. Thus, we successfully obtained double-perovskite-type LLT crystals by TSFZ method. The diffraction peaks of the grown crystals shifted to lower angle than that of the feed rod. It had been reported that the LLT lattice expands as the extra Li evaporation [21]. The peak shift in the grown crystals agrees with the results of the quantitative analysis by EPMA. The diffraction intensity due to the superstructure in the grown crystals was high as compared with the sintered LLT. This indicates that the grown crystals have a high crystallinity.

2. EDX or EPMA compositions (average value and standard deviation) of the successfully grown crystals should be included.

→Thank you for your suggestion. We included the EPMA compositions (average and standard deviation).

We added the following sentences in section 3.3:

The chemical composition of the grown crystals was determined to be $\text{La}_{0.61 \pm 0.02} \text{Li}_{0.17 \pm 0.006} \text{TiO}_3$ ($x=0.057 \pm 0.002$) by quantitative analysis using EPMA. The Li content in the grown crystals was lower than that in the feed. The low Li content is due to evaporation of Li from the molten zone during the crystal growth and a lower distribution coefficient of Li into LLT.

We also added the experimental method of the quantitative analysis in Section “Experimental Procedure” as follows;

Also the concentration of La and Ti in the grown crystals were determined by quantitative analysis using a $\text{La}_2\text{Ti}_2\text{O}_7$ single crystal as a standard sample, and Li concentration $3x$ in the composition $\text{La}_{2/3-x}\text{Li}_{3x}\text{TiO}_3$ was estimated using atomic ratio La/Ti.

3. It would be nice if AC conductivity of oxygen annealed crystals is also included.

→As described the first sentence in section 3.4, the crystals for the measurement were already annealed in oxygen. We added “ The crystals were annealed in oxygen.” also in figure caption of Fig. 5.

4. Habit plane should be defined. It is not clear from the description given in paragraph 1 of section 3.3 as to how the habit plane formation hinders the growth. However, the paragraph 2 is clear where feeding speed is discussed. I will appreciate if paragraph 1 is also made more useful.

→We deleted sentence in paragraph 1 of section 3.3 due to deletion of Fig. 3.

5. Figure 3 is not useful as it doesn't reveal much. You can however choose a picture showing the habit planes. If such a picture is not available you can remove Fig. 3, as you will anyway have one extra figure showing the powder XRD of the crystals.

→We removed Fig. 3. We changed Fig. 4 to Fig. 3. We added XRD pattern of the crystals on Fig.4.

6. No mention of Li evaporation during the growth? In my view, this is an important parameter for reproducibility of these experiments. I suggest that this point be mentioned and discussed.

→We agree with you and have incorporated this suggestion. From the EPMA composition, Li composition in the grown crystals was lower than the starting composition. The low Li content in the grown crystals is due to not only Li evaporation during growth also a low distribution coefficient of Li into LLT. In general, the melt growth causes a decrease of a solute into solid solution in the case of a distribution coefficient of a solute into a solid solution lower than unity. We added data by quantitative

analysis, and mentioned the Li evaporation in section 3.3 as follows.

The chemical composition of the grown crystals was determined to be $\text{La}_{0.61\pm 0.02}\text{Li}_{0.17\pm 0.006}\text{TiO}_3$ ($x=0.057\pm 0.002$) by quantitative analysis using EPMA. The Li content in the grown crystals was lower than that in the feed. The low Li content is due to evaporation of Li from the molten zone during the crystal growth and a lower distribution coefficient of Li into LLT.

Reviewer: 2

Comments to the Author(s)

Dear authors,

This paper describes crystal growth of LLT by TSFZ method. They paid a lot of attention to decomposition of the compound, incongruent melting, to achieve stable growth conditions. As a result, they succeeded to obtain really bulk single crystals. I think this is a very good work in terms of crystal growth.

To make this manuscript more interesting one for the readers, I encourage authors to add some results and descriptions.

Authors mentioned, 'decrease the growth temperature to prevent Li evaporation from the molten zone.' However, they did not disclose the effect of Li loss. It can be possible to estimate Li loss during growth by measuring total weight of feed rod and obtained crystals. I know not only loss of lithium but also oxygen deficiency is another cause of weight loss. However, rough estimation of Li loss may be possible.

→ We thank the referee for fruitful suggestions. We observed Li vaporization during growth because a quartz tube around the growth region becomes white by deposition of Li compound into the quartz tube even though TSFZ growth.

We corrected “prevent” to “reduce” in the sentence which you suggested as follows:

decrease the growth temperature to **reduce** Li evaporation from the molten zone.

We also added the results of the quantitative analysis by EPMA in section 3.3 and discussed about Li vaporization as follows:

The chemical composition of the grown crystals was determined to be $\text{La}_{0.61\pm 0.02}\text{Li}_{0.17\pm 0.006}\text{TiO}_3$ ($x=0.057\pm 0.002$) by quantitative analysis using EPMA. The Li content in the grown crystals was lower than that in the

feed. The low Li content is due to evaporation of Li from the molten zone during the crystal growth and a lower distribution coefficient of Li into LLT.

I suppose Li-loss is also an important fact relating to the authors claim. In order to conclude incongruent melting, it is necessary to show that change of composition in molten zone was not due to loss of lithium but due to incongruent melting. To mention about phase diagram, chemical composition of the system must be kept constant during the process. Loss of lithium is a potential cause for misleading of phase relationships. I do not know if the authors have results to mention about Li loss during growth. However, I would like to ask the author to give quantitative or qualitative estimation of Li-loss during growth process.

→Thank you for your suggestion. We measured chemical compositions of the grown crystals by EPMA. We think that Li evaporate during the crystal growth because Li composition in grown crystals differs from the starting composition. We mentioned the chemical compositions of the grown and Li evaporation in section 3.3.

The conductivity is affected by the number of carrier and mobility. It seems that anisotropy in conduction was more significant than the value reported in a literature. I suppose this is an evidence for high quality of the crystal grown by the authors. However, it is not possible to discuss conductivity in detail only from the results measured at room temperature. To give useful information to readers, it is a good idea to show temperature dependence of conductivity. Evaluation of the activation energy of conduction is necessary for understanding of the conduction mechanism. As the quality of the crystal seems to be sufficiently high to evaluate reliable physical parameters, I encourage the authors to work on temperature dependence measurements.

→We thank you for fruitful suggestions. Although we think that it is a good idea to show temperature dependence of conductivity to give useful information, we can not measure a temperature dependence of conductivity because there is no equipment attached heater. However, we work toward to evaluate the activation energy of conduction in the future work.